# LAMP2A-dependent chaperone-mediated autophagy enhances oxidative stress resistance in gastric cancer cells through selective degradation of accumulated oxidized DJ-1

Shuangshuang Le[1�], Tongtong Guo[2�], Meng Yang[1], Tianjuan Tang[3], Zhijie Liu[4], Yang Zheng[3*], Maogui Pang [3*]

1 Department of Oncology, 904 Hospital of PLA Joint Logistic Support Force, Wuxi, China, 2 School of Life Sciences, Hubei University, Wuhan, Hubei, China, 3 Department of Gastroenterology, 904 Hospital of PLA Joint Logistic Support Force, Wuxi, China, 4 Department of Anesthesiology, 904 Hospital of PLA Joint Logistic Support Force, Wuxi, China

☏ These authors contributed equally to this work.
* pmg575626@163.com (MP); zhengycloud@163.com (YZ)

## Abstract

Chaperone-mediated autophagy (CMA) promotes cancer cell survival by selectively removing oxidatively damaged proteins, yet its precise molecular mechanisms and role in redox adaptation remain incompletely understood. This study aimed to elucidate the function of CMA in regulating oxidative stress resistance in gastric cancer (GC) cells, focusing on the LAMP2A–DJ-1 regulatory axis. LAMP2A expression was assessed in GC tissues and cell lines via immunohistochemistry, qPCR, and western blot. Oxidative stress models were established using hydrogen peroxide ($H_2O_2$). Genetic manipulation of LAMP2A was performed to evaluate its impact on cell proliferation, apoptosis, and CMA substrate recognition. Protein interactions were examined by co-immunoprecipitation and immunofluorescence. We found that LAMP2A was upregulated in GC and further induced by oxidative stress. Knockdown of LAMP2A impaired CMA activity, sensitizing GC cells to $H_2O_2$-induced apoptosis. DJ-1, an antioxidant protein, was identified as a CMA substrate containing a conserved KFERQ-like motif. Oxidative stress enhanced DJ-1–LAMP2A interaction and promoted their lysosomal colocalization. LAMP2A deficiency led to accumulation of hyperoxidized DJ-1, concomitant with upregulation of pro-apoptotic BAX and downregulation of anti-apoptotic Bcl-2. We identify hyperoxidized DJ-1 as a novel CMA substrate and demonstrate that LAMP2A-dependent clearance of oxidized DJ-1 constitutes a key adaptive mechanism that maintains redox homeostasis and promotes survival in gastric cancer cells under oxidative stress.

**Data availability statement:** All relevant data are within the manuscript and its Supporting information files.

**Funding:** This study was financially supported by Wuxi Health and Family Planning Commission in the form of a grant (Q202559) received by MP. This study was also financially supported by Wuxi Municipal Bureau on Science and Technology in the form of a grant (K20252010) received by MP.

**Competing interests:** The authors have declared that no competing interests exist.

**Abbreviations:** ATCC, American Type Culture Collection; BCA, Bicinchoninic acid; CCK-8, Cell Counting K it-8; GC, Gastric cancer; DAPI, 4',6-Diamidino-2-phenylindole; DMEM, Dulbecco's modified Eagle's medium; FBS, Fetal bovine serum; $H_2O_2$, Hydrogen peroxide; NC, Negative control; PBS, Phosphate buffered solution; qPCR, Quantitative polymerase chain reaction; shRNA, Short hairpin RNA; WB, Western blot.

## 1. Introduction

Gastric cancer (GC) ranks fifth in global incidence and third in cancer-related mortality [1]. Although biomarkers such as PD-L1, MSI, and HER2 can guide treatment decisions [2], their limited specificity and sensitivity underscore the urgent need to discover novel molecular markers, which is essential for improving early diagnosis and patient survival. Oxidativestress, characterized by an imbalance between reactive oxygen species (ROS) production and antioxidant defenses plays a dual role in tumor biology [3]. While moderate ROS levels can drive proliferation and survival, excessive accumulation induces cytotoxic damage [4]. The rapid proliferation of tumor cells generates high levels of ROS. However, they can evade senescence and apoptosis by enhancing their intrinsic antioxidant defenses [5].

Autophagy, a conserved lysosomal degradation pathway, is crucial for maintaining cellular homeostasis under stress. Among its three forms—macroautophagy, microautophagy, and chaperone-mediated autophagy (CMA)—CMA stands out for its selectivity [6]. CMA targets cytosolic proteins bearing KFERQ-like motifs, which are recognized by heat shock cognate 70 kDa protein (HSC70) and translocated into lysosomes via the receptor lysosome-associated membrane protein 2A (LAMP2A) for degradation [7,8]. CMA activation has been implicated in promoting tumor progression across various malignancies [9]. For example, in hepatocellular carcinoma, LAMP2A downregulation promotes proliferation and migration via YAP1- and IL6ST-dependent pathways [10]. Conversely, in glioma models, increased CMA activity—achieved through LAMP2A upregulation—similarly enhances proliferation and invasion [11]. The functional importance of CMA in oncogenesis is further underscored by studies in non-small cell lung cancer (NSCLC), where inhibiting the HSC70–LAMP2A interaction blocks CMA and suppresses tumor growth [12]. In breast cancer, CMA promotes cell survival by selectively degrading oxidatively damaged proteins [13], highlighting its critical role in oxidative stress response. Notably, in the context of gastric cancer, LAMP2A—the key receptor in CMA—has been identified not only as a potential early biomarker for gastric mucosal precancerous lesions but also as a specific marker for gastric cancer itself [14]. Therefore, our study will further explore the role and molecular mechanism of CMA in regulating the anti-oxidative stress of gastric cancer cells. We hope establish a mechanistic framework for CMA-mediated oxidative stress adaptation in gastric cancer cells.

## 2. Materials and methods

### 2.1. Tissue microarray analysis

We investigated LAMP2A expression in gastric cancer tissues using a commercial tissue microarray from February 2023 to March 2023. The commercial gastric cancer tissue microarray (Catalog No: HStmA020PG01) was obtained from Shanghai Outdo Biotechnology Co., Ltd. The chip consists of samples from 10 primary gastric cancer tissues, 10 pairs of adjacent nontumor tissues.

All tissues in the commercial tissue microarray were obtained by the supplier with informed consent under ethical guidelines. Thus, analysis of these anonymized samples was deemed non-human subjects research and exempt from approval by our hospital's Ethics Committee.

## 2.2. Cell culture

Normal human gastric mucosal epithelial GES-1 cells and gastric cancer cell lines (AGS, MKN45, HGC27, MKN28) were obtained from the ATCC. Cells were cultured in DMEM (Gibco, 11995040) supplemented with 10% fetal bovine serum (FBS; Gibco, 10099-141) and 1% penicillin-streptomycin (Pen-Strep; ThermoFisher, 15140122), and maintained at 37°C with 5% CO2. All cell lines were validated through short tandem repeat (STR) profiling, with morphological and functional characteristics frequently checked, and confirmed mycoplasma-free.

## 2.3. Cell transfection

The LAMP2A coding sequence synthesized by TsingkeBiotechnology Co.,Ltd was cloned into pcDNA3.1 to generate the LAMP2A-overexpression plasmid. Cells were transfected at 37°C upon reaching 40% confluence. The lentiviral vectors with shRNAs against LAMP2A were purchased from GeneChem Company, Shanghai, China (PIEL248064052). Cell transduction was performed according to the manufacturer's protocol.

## 2.4. RT - qPCR

Total RNA was extracted using an RNA purification kit (Invitrogen, K0731). Reverse transcription was performed with Takara reagents under standard conditions (37°C, 15 min; 85°C, 5 s). PCR amplification was conducted using a two-step cycling protocol (40 cycles): 95°C for 10 s denaturation and 60°C for 30 s extension. *LAMP2A* and *β-actin* primers (Table 1) were custom-synthesized by TsingkeBiotechnology Co., Ltd. RT-qPCR was conducted on a LightCycler480 system (Roche). *LAMP2A* and *β-actin* expression were quantified using the $2^{-\Delta\Delta}$ Ct method.

## 2.5. Cell counting kit-8 assay

Cells were seeded in 96-well plates (2,000 cells/well) and cultured under standard conditions (37°C, 5% $CO_2$). CCK-8 reagent (GLPBIO, GK10001) was added to the medium. After incubating for 2 hours, the absorbance was measured at 450 nm using a microplate reader (Thermo Fisher, VLBL00GD2).

## 2.6. $H_2O_2$-induced oxidative stress model construction

The $H_2O_2$ concentrations were selected based on preliminary CCK-8 assays to identify doses that induce sub-lethal (150 μM) and severe (300 μM) oxidative stress across the four cell lines. For the 5-day prolonged oxidative stress study, we implemented an 8-hour daily $H_2O_2$ (150 μM) exposure protocol rather than continuous stimulation, considering $H_2O_2$'s chemical instability and to preserve cell viability. After each daily treatment, the medium was replaced with standard culture medium for continued cultivation.

**Table 1. Primer and small interfering RNA sequences.**

| |
|---|
| *LAMP2A* forward: 5'- GTGCAACAAAGAGCAGACTGT −3' |
| *LAMP2A* reverse: 5'- CGCTATGGGCACAAGGAAGT −3' |
| *β-actin* forward: 5'- GGACTTCGAGCAAGAGATGG −3' |
| *β-actin* reverse: 5'- AGCACTGTGTTGGCGTACAG −3' |
| *LAMP2A*, lysosome-associated membrane protein 2A |

## 2.7. Immunohistochemical staining

The gastric cancer tissues and adjacent normal tissues were purchased tissue microarrays obtained from Shanghai Outdo Biotechnology Co., Ltd (Product ID: HStmA020PG01). The commercial tissue microarray contains paired primary gastric cancers and adjacent non-tumor tissues from the same patients. Tissue sections underwent sequential processing: baking at 65°C for 3 h, rehydration, EDTA-based antigen retrieval (pH 8), peroxidase blocking with 3% $H_2O_2$, and serum blocking. Primary antibody incubation used rabbit anti-LAMP2A (dilution 1:200, 28477-1-AP, Proteintech) overnight at 4°C, followed by 30 min treatment with HRP-conjugated secondary antibodies (ZSGB). DAB chromogenic reaction was performed for 1 min using a commercial kit. The tissue sections underwent hematoxylin counterstaining, followed by sequential ethanol dehydration, xylene clearing, and final mounting using neutral balsam medium. Whole-slide imaging was conducted on an Olympus VS120 system.

## 2.8. Flow cytometric analysis of apoptosis

Cells were harvested and washed twice with PBS, and resuspended in 1× Binding Buffer to achieve $1 \times 10^6$ cells/mL. 100 μL cell suspension were stained with PE Annexin V and 7-AAD (5 μL each) for 15 min in light-protected conditions.

## 2.9. Immunofluorescence

Cells were seeded in chamber slides at $5 \times 10^4$ cells/well and cultured for 24 h. Following treatment with 1 mL 300 μM $H_2O_2$ for 24 h, cells were fixed with 4% paraformaldehyde (200 μL/well, 15 min RT), then washed twice with PBS (2 min/wash) and thrice with PBS (5 min/wash). Cells were permeabilized with 200 μL of 0.1% Triton X-100 (T8787, Sigma-Aldrich) for 15 min. After three PBS washes (5 min each), nonspecific binding sites were blocked with 200 μL of immunofluorescence blocking buffer (P0102, Beyotime) for 1 h at 25°C. Primary antibodies (1:200; ab199337, Abcam) were applied (200 μL/chamber) and incubated overnight at 4°C. After three PBS washes, samples were incubated with species-specific Alexa Fluor-conjugated secondary antibodies (1:500; A-11034, Thermo Fisher) for 1 h at 25°C with gentle agitation, protected from light. To control for non-specific binding of the secondary antibody, a negative control omitting the primary antibody was included in each experiment. Nuclei were counterstained with DAPI (40728ES03, Yeasen) for 30 sec. Fluorescence imaging was performed using a Nikon A1R HD25 confocal system (AX/AXR with N SPARC). Protein localization analysis included systematic random sampling of ≥5 fields per condition, with signal specificity confirmed through negative controls lacking primary antibodies.

## 2.10. Western blotting

Following 72-hour treatment, cells were harvested and lysed in ice-cold RIPA buffer. Lysates were centrifuged at 12,000 ×g for 15 min at 4°C, and supernatants were collected. Protein concentrations were quantified using a BCA assay (23235, Thermo Fisher). Protein samples were denatured at 95°C for 5 min in a preheated thermal cycler. Electrophoresis was performed using 10% SDS-PAGE gels (4561033, Bio-Rad) at 100 V for 120 min. Resolved proteins were transferred to membranes (0.45 μm, 1620115, Bio-Rad) via wet transfer (30 V, 60 min, 4°C). Membranes were blocked with 5% non-fat milk for 1 h. Primary antibodies incubated overnight at 4°C: LAMP2A (1:1000, #Ab125068, Abcam), Dj-1(1:1000, # Ab76008, Abcam), Bcl-2 (1:1000, #4223, CST), Bax (1:1000, #5023, CST), and β-actin (1:1000, # Ab5694, Abcam). After three 10-min TBST washes, membranes were incubated with species-matched HRP-conjugated secondary antibodies (anti-rabbit: 1:2,000, # Ab7074, Abcam; anti-mouse: 1:2,000, # Ab7076, Abcam) for 1 h at 25°C. Membranes underwent three 5-min TBST washes before chemiluminescent detection using ECL Prime substrate (WBKLS0500; Millipore) with 5-min substrate incubation. β-actin served as loading control. Protein signals were digitally captured using a ChemiDoc XRS+ system (12003154, Bio-Rad). The original, uncropped Western Blot images supporting all figures are provided in S1 Raw Images.

## 2.11. Co-immunoprecipitation analysis

Cells were collected and lysed in 1 mL of IP lysis buffer. After full lysis, centrifuge at 4 degrees and 12,000g for 15 minutes to obtain total cell protein as the input group. Use 20μl magnetic beads (#P2151, BeyoMag, Inc.) to immuno-precipitate the lysate and incubate with anti-DJ-1 antibody (1:1000, #Ab76008, Abcam) at 4°C overnight. As a negative control, normal rabbit IgG (1:1000, #2729, Cell Signaling Technology) was used in place of the primary antibody under identical conditions to confirm the specificity of the antibody-protein interaction. Samples were centrifuged (250×g, 5 min, 4°C), and supernatants removed via vacuum aspiration. Magnetic beads were washed thrice with ice-cold PBS. Following three PBS washes, immunocomplex-bound beads were resuspended in 60 μL IP lysis buffer combined with 15 μL 2× Laemmli sample buffer, then thermally denatured at 100°C for 10 min using a programmable heating block (ThermoFisher, 88870001). The supernatant is transferred to a new centrifuge tube, and the resulting product is the IP product, which is stored at -20 degrees. Western blot assay was performed using an anti-LAMP2A antibody (dilution 1:1000, # Ab125068, Abcam).

## 2.12. Statistical methods

Statistical analyses were performed using SPSS Statistics 23.0 (v23.0.0; IBM). For two-group comparisons, independent two-sample Student's t-tests with Welch's correction were applied. Multi-group analyses employed one-way ANOVA followed by Dunnett's post hoc test for comparisons against a specified control group. *$p < 0.05$ indicates statistical significance. All raw numerical data underlying the figures are provided in S2 Raw Data

## 3. Results

### 3.1. Upregulation of LAMP2A in gastric cancer promotes CMA

First, we validated the expression of LAMP2A in GC using immunohistochemistry in gastric tumors and normal tissue (Fig 1A). Then, we analyzed the transcript levels of LAMP2A in paired gastric adenocarcinoma and adjacent normal mucosa from the TCGA and GEO databases (Fig 1B). Four gastric adenocarcinoma cell lines and non-neoplastic gastric epithelial cells were analyzed. LAMP2A mRNA expression was quantified by RT-PCR (Fig 1C). Protein levels were assessed via immunoblotting WB (Fig 1D).

LAMP2A is the rate-limiting component of chaperone-mediated autophagy (CMA), which directly governs lysosomal substrate flux through its assembly into multimeric translocation complexes. Up-regulated and down-regulated expression of LAMP2A can enhance and decrease the activity of CMA, respectively [15]. Consistent with previous reports [14], we confirmed that LAMP2A is consistently upregulated in gastric adenocarcinoma tissues and cell lines. (Fig 1A-D). Given that LAMP2A is the rate-limiting factor of CMA, its elevated level may indicate enhanced CMA activity in gastric cancer cells compared to normal gastric mucosa cells.

### 3.2. Oxidative stress up-regulated the expression of LAMP2A in gastric cancer cells

Experimental evidence indicates that oxidative stress induces LAMP2A upregulation in glioblastoma cells [16]. Hydrogen peroxide ($H_2O_2$) treatment has been established as a standard method for generating experimental models of oxidative stress [17]. To investigate oxidative stress as an inducer of LAMP2A upregulation in gastric cancer, four cell lines were exposed to $H_2O_2$ at two concentrations of 150μM and 300μM, respectively, to establish a cell model of severe oxidative stress. Western blot analysis demonstrated $H_2O_2$-induced elevation of LAMP2A protein levels. These results suggest that oxidative stress is a driver of LAMP2A upregulation in GC (Fig 2). These $H_2O_2$-induced upregulations of LAMP2A were effectively prevented by pretreatment with the antioxidant N-acetylcysteine (NAC) (S3 Fig). This provides direct evidence that the observed effects are specifically mediated by ROS generated from $H_2O_2$.

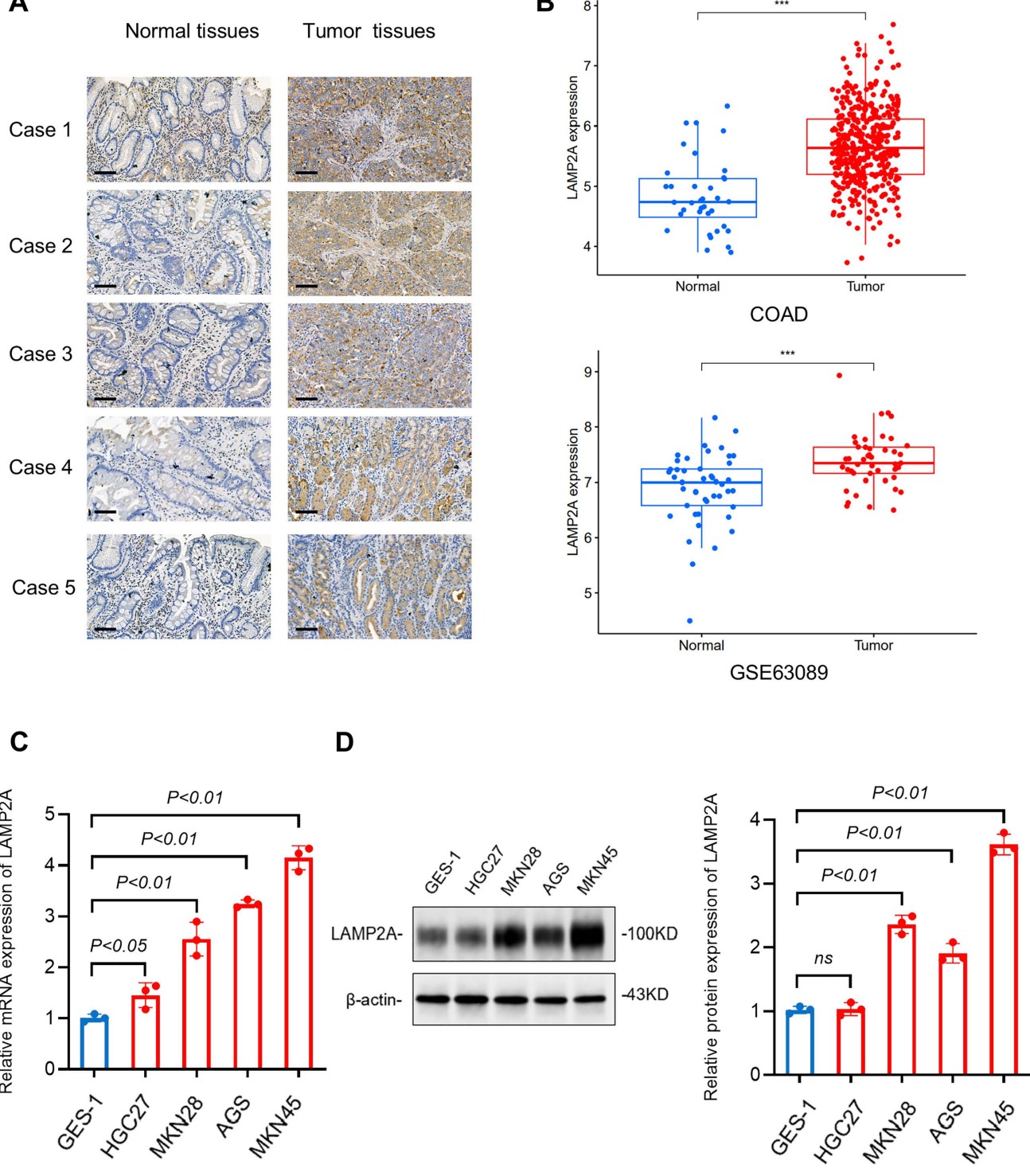

**Fig 1. Detection of LAMP2A expression in gastric cancer.** (A) Immunohistochemical staining of LAMP2A in gastric tumor and normal tissue. (Scale bar: 20 μm). (B) The TCGA and GEO databases show the mRNA expression levels of LAMP2A in gastric tumors and normal tissue. (C)The qPCR results showed the LAMP2A expression at mRNA levels. (D) Western blot analysis of the level of LAMP2A in several cell lines. Note:TCGA(https://portal. gdc.cancer.gov/)GEO(https://www.ncbi.nlm.nih.gov/geo/query/acc.cgi?acc=GSE63089). Data are presented as mean±SD of three independent experiments. ns, no significance, *P<0.05. **P<0.01. ***P<0.001.

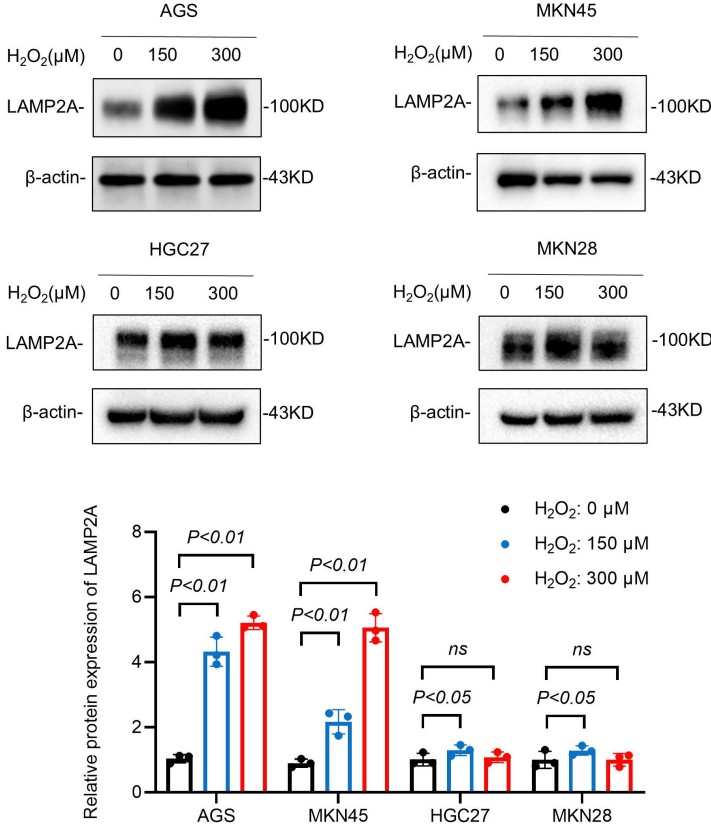

**Fig 2. Oxidative stress upregulates the expression of LAMP2A in cell lines.** Four cell lines were exposed to $H_2O_2$ (150μM, 300μM), respectively, to establish a severe oxidative stress cell model. WB results showed the protein expression of LAMP2A. Data are presented as mean±SD of three independent experiments. ns, no significance, *P < 0.05. **P < 0.01.

### 3.3. Construction of LAMP2A knockdown and overexpression cell models

AGS cells were transfected with a recombinant plasmid encoding LAMP2A to establish stable overexpression, with experimental groups designated as: negative control (NC) and LAMP2A-overexpressing (LAMP2A). MKN45 cells transduced with LAMP2A-targeted shRNA lentiviral particles to create knockdown models, with groups designated as: negative control (LV-sh NC) and LAMP2A-knockdown (LV-sh L2A).

RT-qPCR showed a significant decrease of LAMP2A in MKN45 cells compared to the LV-sh NC control (p<0.01, Fig 3A). LAMP2A mRNA expression was significantly elevated in AGS cells compared with negative control (NC) cells (**p < 0.01; Fig 3C). Western blot analysis confirmed successful LAMP2A knockdown and overexpression (*p < 0.05, **p < 0.01; Fig 3B, D). We selected LV-sh LAMP2A-2 for constructing the LAMP2A knockdown model due to its optimal knockdown efficacy (Fig 3B).

### 3.4. Loss of LAMP2A promoted oxidative stress-induced apoptosis of gastric cancer cells

To investigate the functional role of LAMP2A in oxidative stress response, we used the previously constructed LAMP2A cell model. CCK-8 analysis revealed that LAMP2A knockdown significantly impaired cellular proliferation both under basal conditions (Fig 4A) and upon stimulation with 150 μM $H_2O_2$. We also demonstrated a significant amplification of proliferation rate disparity between LAMP2A and control groups under $H_2O_2$-induced oxidative stress relative to basal

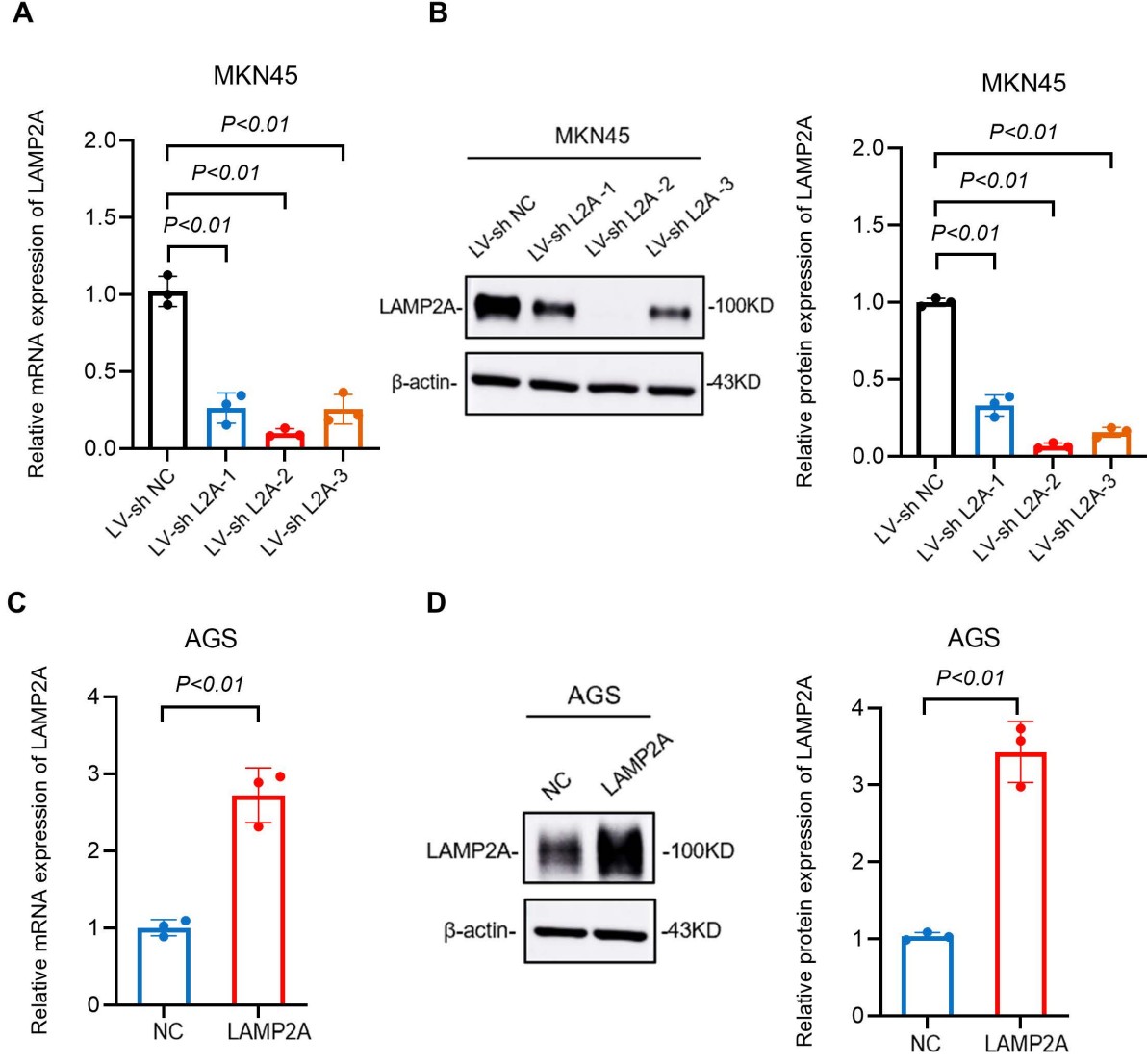

**Fig 3. Validation of LAMP2A knockdown and overexpression efficiency in cellular models through qPCR and WB.** (A, B) Verification of LAMP2A knockdown efficiency in MKN45 cell. (C, D) Verification of LAMP2A overexpression efficiency in AGS cell. Note: NC, negative control. Data are presented as mean±SD of three independent experiments. ns, no significance, *P<0.05. **P<0.01.

culture conditions (Fig 4B). We also conducted CCK-8 assays using both AGS cells and the AGS line stably overexpressing LAMP2A (Fig 4C). Under treatment with 150 µM $H_2O_2$, AGS cells overexpressing LAMP2A exhibited a markedly higher proliferation rate compared to the negative control (NC) cells (Fig 4D). Flow cytometric analysis revealed that LAMP2A-knockdown significantly increased the proportion of cells in early apoptosis (Annexin V+/7-AAD-) upon $H_2O_2$ exposure, which was accompanied by a subsequent rise in late apoptotic/necrotic cells (Annexin V+/7-AAD+), leading to a dramatic increase in the total apoptosis rate (Fig 4E). We also confirm that knockdown with LV-shL2A-3 similarly sensitizes MKN45 cells to $H_2O_2$-induced apoptosis, replicating the core phenotypes observed with LV-shL2A-2 (S4 Fig). Conversely, LAMP2A-overexpression notably reduced the early apoptotic population under oxidative stress, demonstrating a protective effect against apoptosis initiation. Its total apoptosis rate was still significantly lower than the control (Fig 4F).

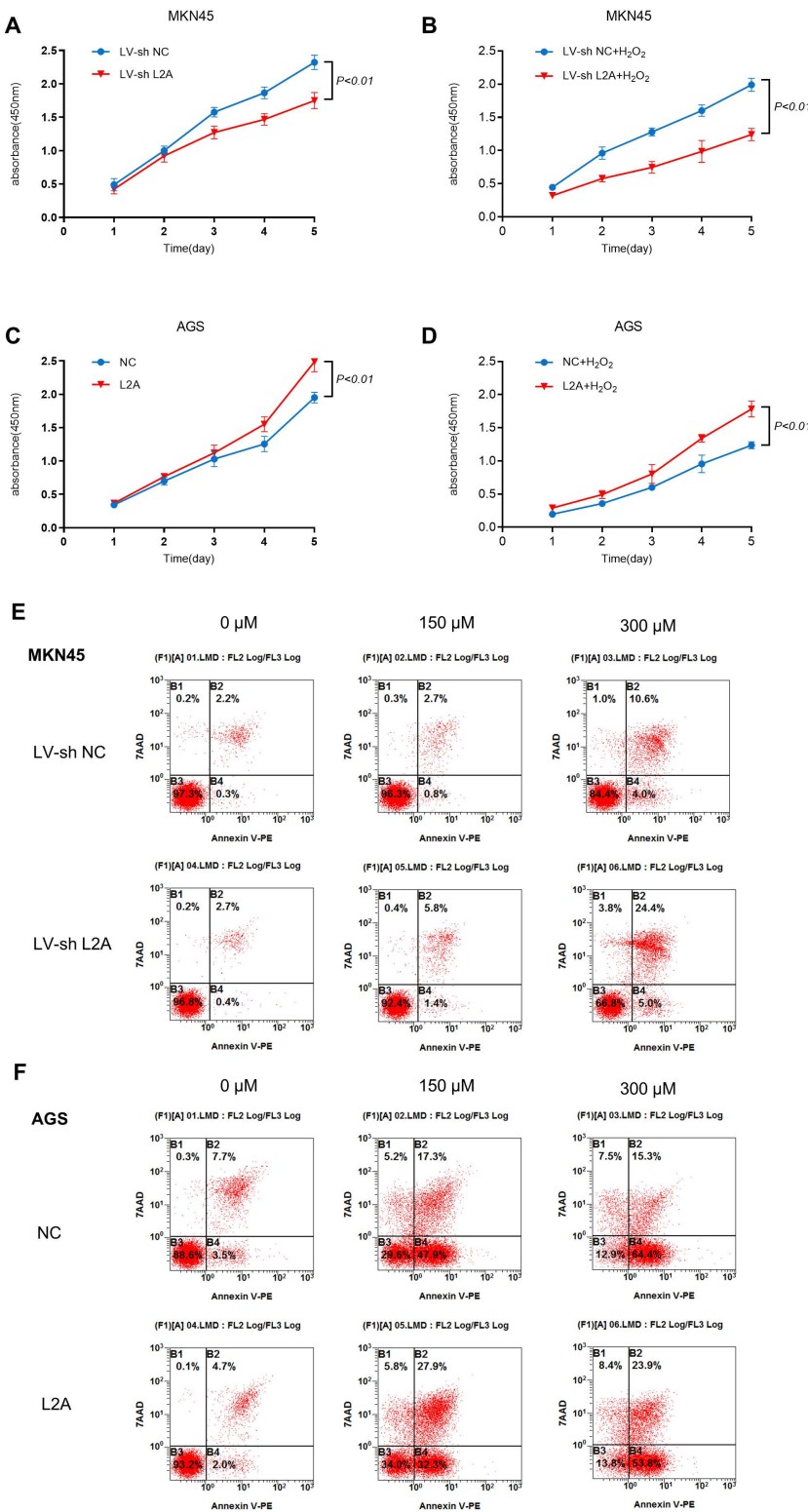

**Fig 4. LAMP2A knockdown promotes apoptosis of gastric cancer cells induced by severe oxidative stress.** (A): Proliferation of the control group and LAMP2A-knockdown group. (B): Proliferation of the control group and LAMP2A-knockdown group following 150 μM H$_2$O$_2$ treatment. (C):

Proliferation of the control group and LAMP2A-overexpress group. (D): Proliferation of the control group and LAMP2A-overexpress group following 150 μM H2O2 treatment. (E): LAMP2A knockdown cell lines were exposed to $H_2O_2$ (0, 150μM and 300μM), and the apoptosis rate was measured by FACS after 24-hour treatment. (F): LAMP2A over-expressing cells were exposed to $H_2O_2$ (0, 150μM and 300μM), and the apoptosis rate was measured by FACS after 24-hour treatment. Note: L2A, LAMP2A; NC, negative control. Data are presented as mean±SD of three independent experiments. ns, no significance, *P<0.05. **P<0.01.

LAMP2A knockdown significantly impaired cellular proliferative capacity and augmented apoptosis under oxidative stress. Since LAMP2A is the limiting receptor for CMA, its knockdown is expected to, and has been widely demonstrated to, inhibit CMA activity [15,19]. These findings demonstrate that LAMP2A knockdown inhibit CMA activity, thereby sensitizing gastric cancer cells to oxidative stress-induced apoptosis.

### 3.5. LAMP2A recognizes DJ-1 as a chaperone-mediated autophagy substrate through direct interaction

Having established the upregulation of the CMA rate-limiting component LAMP2A in gastric cancer, we sought to identify specific CMA substrates that contribute to oxidative stress resistance. We focused on the antioxidant protein DJ-1 (PARK7), a redox-sensitive protein implicated in cancer and neurodegeneration [18], which may be a substrate for CMA [14]. Structural analysis reveals that DJ-1 contains an evolutionarily conserved KFERQ-like motif (Fig 5A), establishing it as a CMA substrate through this recognition signature, the canonical recognition signature for CMA substrates [19].

WB revealed elevated DJ-1 protein levels in gastric cancer cell lines, with further upregulation observed under hydrogen peroxide treatment (Fig 5B, C). To investigate DJ-1's role in LAMP2A-mediated (CMA) under oxidative stress conditions, we assessed LAMP2A-DJ-1 interactions via co-immunoprecipitation (Co-IP) and immunofluorescence (IF), comparing basal and $H_2O_2$-treated conditions. Co-IP results demonstrated enhanced DJ-1–LAMP2A binding following 24-hour $H_2O_2$(300μM) treatment compared to untreated controls (Fig 5D). Immunofluorescence imaging confirmed marked colocalization of LAMP2A and DJ-1 under oxidative stress versus basal conditions (Fig 5E).

### 3.6. LAMP2A knockdown promotes DJ-1-induced apoptosis of gastric cancer cells under severe oxidative stress

To explore whether LAMP2A could regulate the expression of DJ-1, we constructed the LAMP2A knockdown gastric cancer cell line MKN45shL2A and its control cell line MKN45shNC, treated gastric cancer cells with hydrogen peroxide (0, 300μM) for 24 hours, and detected the changes of DJ-1 after LAMP2A knockdown by WB. The results showed that LAMP2A knockdown increased DJ-1 expression regardless of $H_2O_2$ stimulation. WB analysis demonstrated that LAMP2A knockdown-mediated DJ-1 upregulation significantly elevated pro-apoptotic BAX expression while suppressing anti-apoptotic Bcl-2 levels, indicative of a pro-apoptotic shift in cellular signaling (Fig 6A). Then, we constructed a shRNA-resistant LAMP2A plasmid and co-transfected it into our LAMP2A-knockdown MKN45 cells. The results (Fig 6B) demonstrate that re-expression of LAMP2A significantly reversed the increased DJ-1and the dysregulation of BAX/Bcl-2 caused by LAMP2A knockdown under oxidative stress (H2O2: 300μM).

## 4. Discussion

Autophagy, a conserved lysosomal degradation pathway, preserves cellular homeostasis by selectively removing damaged organelles and misfolded proteins. There are three types of autophagy, i.e., macroautophagy, chaperone-mediated autophagy (CMA), and microautophagy. CMA is mechanistically distinct from macroautophagy and microautophagy. Substrate proteins are selectively recognized by cytosolic chaperones, directly translocated across the lysosomal membrane via LAMP2A receptors, and degraded without intermediate vesicle encapsulation [20].

CMA is the first autophagy discovered to be selective for protein degradation. It regulates cell function by specifically recognizing an amino acid sequence "KFERQ" in substrate proteins, thereby selectively degrading key intracellular proteins [21]. Emerging evidence indicates that CMA is activated across multiple tumor types. Furthermore, stressors

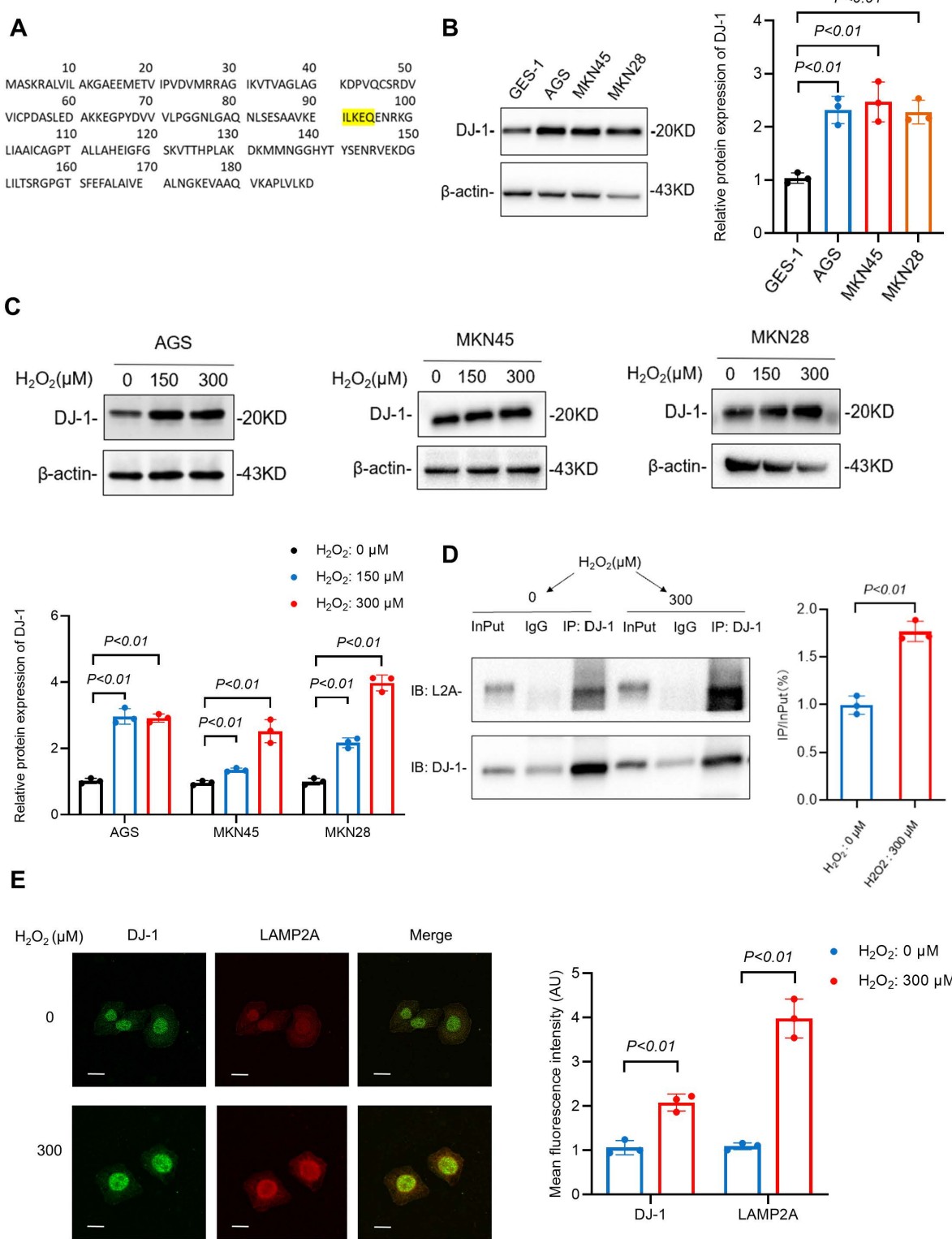

**Fig 5. DJ-1 is the substrate for CMA, and LAMP2A directly interacts with DJ-1.** (A): Identification of a conserved KFERQ-like motif in DJ-1, a CMA substrate recognition signature. (B): WB analysis of DJ-1 protein expression levels. (C) $H_2O_2$ dose-dependent upregulation of DJ-1 protein in gastric

cancer cells (150 µM, 300 µM; 24 h treatment). (D): Co-IP confirms enhanced DJ-1/LAMP2A interaction in $H_2O_2$-treated (300 µM, 24 h) MKN45 cancer cells. (E): IF demonstrates oxidative stress-induced colocalization of DJ-1 and LAMP2A in MKN45 cells (300 µM $H_2O_2$, 24 h). Data are presented as mean±SD of three independent experiments. ns, no significance, *P < 0.05. **P < 0.01.

such as nutrient deprivation, hypoxia, and ROS have been demonstrated to serve as potent inducers of CMA activity [22]. Therefore, investigating the relationship between CMA and oxidative stress resistance in tumor cells, and elucidating the underlying molecular mechanisms, holds significant research value.

LAMP2A is a key receptor molecule in the CMA process. Substrate proteins require binding to LAMP2A on the lysosomal membrane for translocation into the lysosomal lumen and subsequent degradation. Increased LAMP2A expression enhances chaperone-mediated autophagy (CMA) activity, with its protein abundance serving as a direct quantitative indicator of CMA functionality [23]. Previous studies demonstrated that the expression of LAMP2A was significantly elevated in gastric cancer [14], indicating that the basic activity of CMA in tumors was enhanced. It has been reported that ROS can upregulate the level of LAMP2A in glioblastoma [10]. Building on the observed LAMP2A upregulation in gastric malignancies, we hypothesized its potential role in mediating oxidative stress adaptation. To test this, four established gastric adenocarcinoma cell lines (AGS, MKN45, HGC27, MKN28) were exposed to $H_2O_2$ gradients for 24h. WB analysis demonstrated dose-dependent elevation of LAMP2A protein abundance under oxidative stress conditions (Fig 2). These findings establish oxidative stress as a potent inducer of LAMP2A expression in gastric cancer cells. To delineate the mechanistic role of LAMP2A in oxidative stress adaptation, we establish oxidative stress models, gastric adenocarcinoma cells were exposed to $H_2O_2$ gradients. LAMP2A-knockdown and LAMP2A-overexpression cell lines were generated. Cellular responses were quantified using CCK-8 proliferation assays and FACS. The results showed that loss of LAMP2A inhibited CMA activity and increased the sensitivity of gastric cancer cells to oxidative stress.

Studies have reported that the oxidative stress regulatory protein DJ-1 (PARK7) may bind to LAMP2A and HSC70, suggesting that they are substrates of CMA [14]. DJ-1 was first discovered in 1997 as an oncogene and was associated with early-onset PD in 2003 [24]. Since then, DJ-1, as an antioxidant stress protein, has attracted increasing attention. DJ-1 can eliminate intracellular ROS through self-oxidation. DJ-1 can activate intracellular multiple molecular signaling pathways and enhance the expression of cellular antioxidant enzymes and promote the elimination of ROS [25]. Tumor cells can also use these antioxidant mechanisms to protect themselves from damage by ROS. DJ-1 is elevated in many malignant tumors and promotes tumor progression. For example, COE-mediated downregulation of DJ-1 may be the primary cause of mitochondrial structural and functional dysfunction in NSCLC, eventually leading to ROS accumulation [26]. The anti-cancer effects of CPX in HCC cells were also attributed to CPX-triggered ROS accumulation and DJ-1 downregulation [27]. In pancreatic cancer, papillary thyroid cancer, and osteosarcoma, DJ-1 knockout inhibits tumor proliferation and promotes apoptosis [28,29]. Emerging evidence in gastric carcinogenesis demonstrates DJ-1 promotes gastric cancer resistance, participates in metastasis, and is associated with poor prognosis [30,31]. Collectively, these findings establish DJ-1 as a critical cytoprotective molecule in tumor cells.

We initially demonstrated that DJ-1 protein expression levels were elevated in gastric cancer cells following oxidative stress induction (Fig 5C), showing its cytoprotective function through redox regulation. However, Co-IP assays revealed enhanced DJ-1/LAMP2A complex formation under oxidative stress relative to basal conditions (Fig 5D). IF analysis further demonstrated pronounced subcellular co-localization of these proteins during severe oxidative stress (Fig 5E). These findings indicate selective degradation of a DJ-1 subpopulation via CMA. Although DJ-1 is well-known for protecting cells by maintaining redox balance and inhibiting apoptosis, we found that CMA, another cytoprotective pathway, targets DJ-1 for degradation. This presents an apparent paradox.

Accordingly, we have undertaken research to address this paradox. Substantial studies have confirmed that Cys106 is an essential residue for DJ-1's antioxidative protective function. Mild oxidation of Cys106 to the sulfinic state ($SO_2H$)

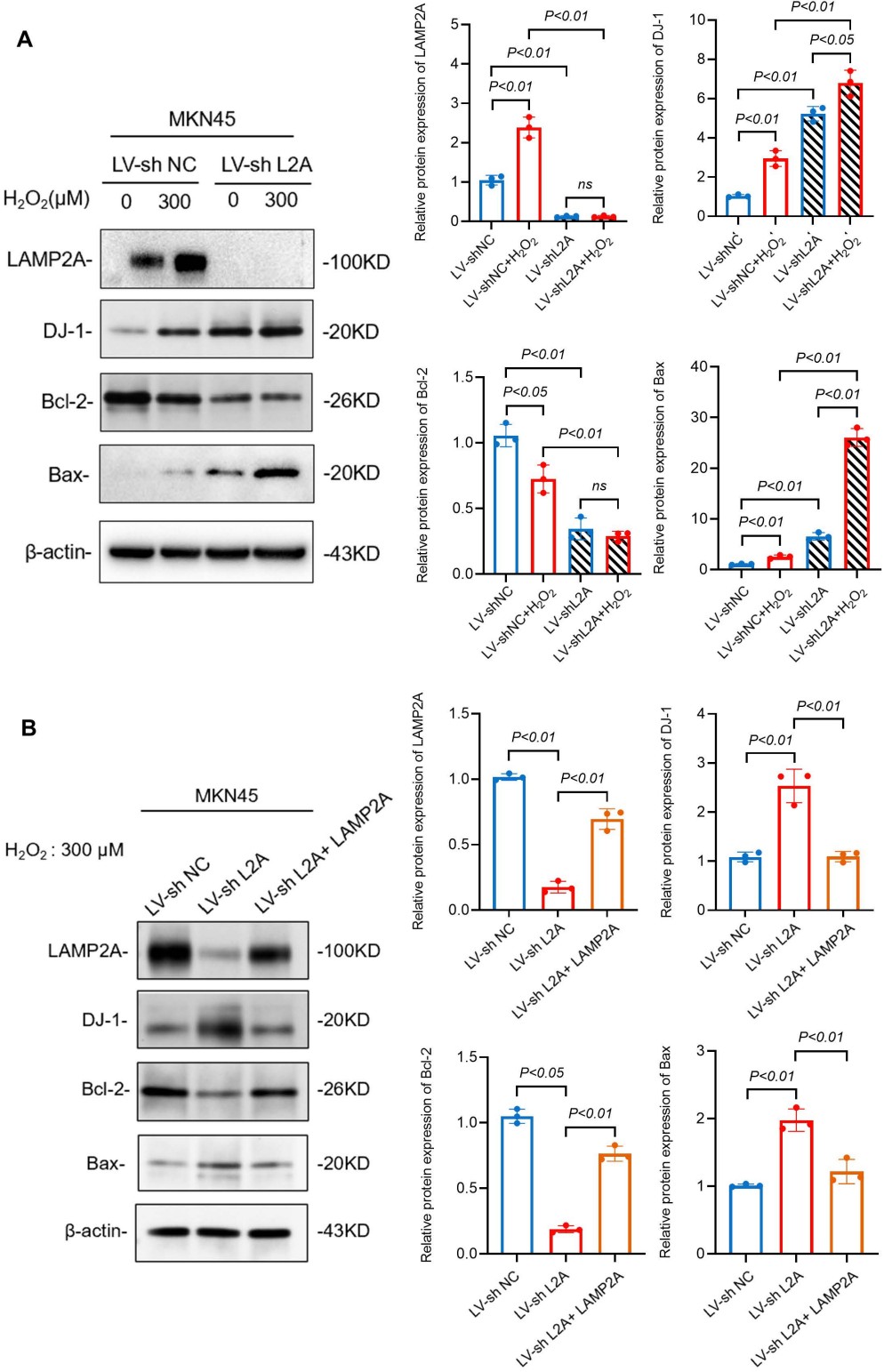

**Fig 6. LAMP2A knockdown promotes DJ-1-induced apoptosis of gastric cancer cells under severe oxidative stress.** (A) MKN45shL2A and MKN45shNC were stimulated by $H_2O_2$ (0,300 μm) for 24 hours, and the protein levels of LAMP2A, DJ-1, apoptosis-related proteins Bcl-2 and BAX were detected by WB. (B) The MKN45shL2A cells were transfected with the LAMP2A plasmid to generate the rescue group (sh L2A+LAMP2A). The

protein levels of LAMP2A, DJ-1, Bcl-2, and BAX were analyzed by WB in all cell groups after a 24-hour exposure to 300 μM $H_2O_2$. Data are presented as mean±SD of three independent experiments. ns, no significance, *$P < 0.05$. **$P < 0.01$.

represents a critical modification for DJ-1 to exert its protective role. Conversely, further oxidation converting Cys106 to the sulfonic acid form ($SO_3H$) promotes DJ-1 aggregation and consequent loss of function [32]. Notably, the presence of hyperoxidized DJ-1 has been detected in the brain tissues of patients with Parkinson's disease and Alzheimer's disease, suggesting that impaired clearance of overoxidized DJ-1 may contribute to cytotoxic damage [33]. Some studies have found that in tumor cells, physiological oxidation enables DJ-1 to sequester ASK1 through direct protein-protein interaction, thereby suppressing JNK/p38 signaling cascades while potentiating cytoprotective autophagy flux. Conversely, pathological hyperoxidation induces DJ-1-ASK1 complex dissociation, liberating ASK1 to phosphorylate downstream effectors including p38 MAPK that ultimately execute apoptosis commitment [34]. This suggests that the hyperoxidized form of DJ-1 is cytotoxic, and timely clearance of this dysfunctional protein is crucial for cell survival. Meanwhile, a study showed that CMA mediated the lysosome-dependent degradation of PARK7. Importantly, CMA preferentially removed the oxidatively damaged nonfunctional PARK7 protein [35].

Therefore, we propose a cytoprotective model in which oxidative stress promotes the DJ-1/LAMP2A interaction. This facilitates CMA-mediated degradation of hyperoxidized DJ-1, thereby maintaining a pool of functional, redox-competent DJ-1 and enabling cancer cells to evade apoptosis. We verified this hypothesis by using the constructed LAMP2A knockdown cell model, and the results showed that down-regulation of LAMP2A could increase the protein level of DJ-1. In addition, we also found that LAMP2A depletion precipitated hyperoxidized DJ-1 accumulation concomitant with apoptotic priming, evidenced by reciprocal BAX upregulation and Bcl-2 downregulation (Fig 6). These findings demonstrate that DJ-1 is degraded via CMA, supporting the conclusion that hyperoxidized DJ-1 serves as a CMA substrate (Fig 7).

Under severe oxidative stress induced by $H_2O_2$, DJ-1 undergoes hyperoxidation. Knockdown of LAMP2A under these conditions impairs CMA-mediated clearance of hyperoxidized DJ-1, leading to its accumulation. Notably, in the absence of $H_2O_2$ stimulation, we also found that LAMP2A knockdown induced DJ-1 accumulation. Regarding this phenomenon, we believe that under physiological conditions, CMA operates as a housekeeping mechanism that sustains proteostasis through continuous clearance of endogenously generated hyperoxidized DJ-1, thereby maintaining redox equilibrium under physiological conditions. However, whether DJ-1 is degraded in a CMA-dependent manner remains controversial due to a lack of conclusive experimental evidence. Our findings in gastric cancer models provide experimental evidence supporting the classification of overoxidized DJ-1 as a CMA substrate in tumors. A key methodological constraint in this study involves our inability to directly interrogate the oxidation state of DJ-1. We acknowledge that the designation of hyperoxidized DJ-1 as a CMA substrate is based on indirect evidence. Directly identifying the targeted oxidized species represents an essential goal for future work, potentially through techniques like redox-specific 2D gel electrophoresis or mass spectrometry. Thus, the mechanistic role of CMA in counteracting oxidative stress during gastric carcinogenesis requires systematic investigation. Furthermore, more studies are needed to definitively establish DJ-1 hyperoxidation as a biomarker of CMA substrate specificity in malignant contexts.

## 5. Conclusions

Our study found that the expression of LAMP2A and DJ-1 is elevated in gastric cancer cell lines compared to normal gastric mucosal cells, and that oxidative stress acts as an inducer of their upregulation. Mechanistically, we found hyperoxidized DJ-1 as a chaperone-mediated autophagy (CMA) substrate, demonstrating that CMA-mediated clearance of oxidized DJ-1 enhances cellular antioxidant capacity and promotes gastric cancer cell survival. These findings elucidate the mechanistic role of chaperone-mediated autophagy (CMA) in regulating tumor cell redox homeostasis. Our work elucidates a key cytoprotective mechanism in gastric cancer, which may provide a basis for novel therapeutic strategies targeting CMA.

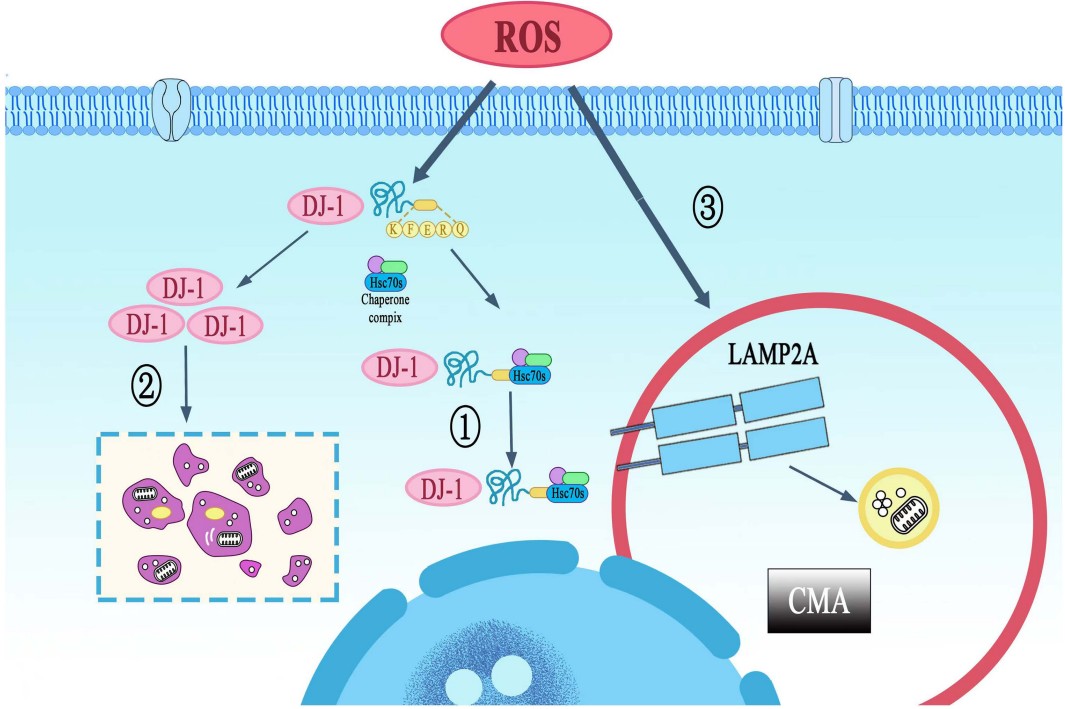

**Fig 7. Mechanisms of LAMP2A-dependent CMA enhances oxidative stress resistance in gastric cancer cells.** ①CMA inhibits tumor cell apoptosis through selective removal of overoxidized DJ-1 protein. ②In pathological conditions, ROS-mediated overoxidation converts DJ-1 into a pro-apoptotic form that triggers gastric cancer cell death, distinct from its physiological oxidative modification under basal oxidative stress. ③Oxidative stress serves as the primary inducer of LAMP2A upregulation in gastric cancer cells, establishing a compensatory mechanism for CMA activation.

## Supporting information

**S1 Raw Western Blot Images. This file contains uncropped, full-length Western blot membranes for all figures.**
(PDF)

**S2 Raw Data. This file includes the numerical raw data underlying all quantitative results.**
(XLSX)

**S3 Fig. The antioxidant N-acetylcysteine (NAC) prevents $H_2O_2$-induced upregulation of LAMP2A and DJ-1, confirming ROS-dependency of the pathway.** (A)Western blot analysis shows protein levels of LAMP2A and DJ-1 in gastric cancer cells pretreated with or without NAC (5 mM, 2 h) followed by exposure to $H_2O_2$ (300 μM, 24 h). Data are presented as mean±SD of three independent experiments. ns, no significance, *$P < 0.05$. **$P < 0.01$.
(TIF)

**S4 Fig. Validation with a second independent LAMP2A-targeting shRNA (LV-shL2A-3). confirms LAMP2A-specific phenotypes.** (A) CCK-8 proliferation assay of LV-shL2A-3 and LV-shNC cells. (B) Proliferation of LV-shL2A-3 and LV-shNC cells after treatment with 150 μM $H_2O_2$. (C) Flow cytometry analysis of apoptosis in LV-shL2A-3 and LV-shNC cells following exposure to $H_2O_2$ (0, 300 μM) for 24 hours. Consistent with the results obtained with LV-shL2A-2, cells transfected with LV-shL2A-3 exhibited significantly increased sensitivity to $H_2O_2$-induced oxidative stress. Data are presented as mean±SD of three independent experiments. ns, no significance, *$P < 0.05$. **$P < 0.01$.
(TIF)

## Author contributions

**Conceptualization:** Shuangshuang Le, Maogui Pang.

**Data curation:** Meng Yang, Tianjuan Tang.

**Formal analysis:** Zhijie Liu.

**Funding acquisition:** Yang Zheng.

**Investigation:** Shuangshuang Le, Tongtong Guo.

**Methodology:** Shuangshuang Le, Tongtong Guo, Maogui Pang.

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
