## [Decision Letter · Decision Letter 0]

29 Sep 2025

PONE-D-25-43608LAMP2A-dependent chaperone-mediated autophagy enhances oxidative stress resistance in gastric cancer cells through selective degradation of accumulated oxidized DJ-1PLOS ONE

Dear Dr. pang,

Thank you for submitting your manuscript to PLOS ONE. After careful consideration, we feel that it has merit but does not fully meet PLOS ONE’s publication criteria as it currently stands. Therefore, we invite you to submit a revised version of the manuscript that addresses the points raised during the review process.

Please find attached the detailed comments from the reviewers. We kindly ask you to carefully address each point raised in your revision. When submitting the revised manuscript, please also provide a point-by-point response to the reviewers’ comments, outlining the changes made or explaining your reasoning if any suggestions were not incorporated.

We look forward to receiving your revised manuscript.

Kind regards,

Zu Ye, Ph.D.

Academic Editor

PLOS ONE

Journal Requirements:

Please confirm at this time whether or not your submission contains all raw data required to replicate the results of your study. Authors must share the “minimal data set” for their submission. PLOS defines the minimal data set to consist of the data required to replicate all study findings reported in the article, as well as related metadata and methods (https://journals.plos.org/plosone/s/data-availability#loc-minimal-data-set-definition)

Reviewers' comments:

Reviewer's Responses to Questions

**Comments to the Author**

1. Is the manuscript technically sound, and do the data support the conclusions?

Reviewer #1: Partly

Reviewer #2: Partly

Reviewer #3: Partly

Reviewer #4: Partly

2. Has the statistical analysis been performed appropriately and rigorously? 

Reviewer #1: Yes

Reviewer #2: Yes

Reviewer #3: I Don't Know

Reviewer #4: Yes

3. Have the authors made all data underlying the findings in their manuscript fully available?

Reviewer #1: No

Reviewer #2: Yes

Reviewer #3: Yes

Reviewer #4: Yes

4. Is the manuscript presented in an intelligible fashion and written in standard English?

Reviewer #1: Yes

Reviewer #2: Yes

Reviewer #3: Yes

Reviewer #4: Yes

5. Review Comments to the Author

Reviewer #1: The manuscript examines the role of chaperone-mediated autophagy (CMA) in gastric cancer under conditions of oxidative stress. The authors demonstrate that LAMP2A, the rate-limiting component of CMA, is upregulated in gastric cancer tissues and further induced upon exposure to H₂O₂. Knockdown of LAMP2A sensitises gastric cancer cells to apoptosis triggered by oxidative stress, whereas its overexpression promotes cell survival. They propose that CMA selectively degrades hyperoxidised DJ-1, thereby maintaining redox homeostasis and protecting tumour cells. The study concludes that the interaction between LAMP2A and DJ-1 represents a key mechanism conferring resistance to oxidative stress in gastric cancer cells.

Major comments:

1. The central claim that hyperoxidised DJ-1 is a CMA substrate is not fully substantiated; the oxidation status of DJ-1 was inferred indirectly and not directly measured (for example, using specific antibodies, redox-sensitive mutants, or mass spectrometry).

2. Rescue experiments are lacking. Demonstrating that shRNA-resistant LAMP2A can reverse the observed phenotypes would considerably strengthen the conclusions.

3. Controls using antioxidants (e.g., NAC, GSH donors) are not included, making it unclear whether the observed effects are truly ROS-dependent.

4. The authors equate LAMP2A knockdown with CMA inhibition, but more direct assessment of CMA activity is necessary (for example, KFERQ reporter assays or lysosomal uptake assays).

Minor comments:

1. The manuscript would benefit from language editing to reduce repetition and improve clarity.

2. Figure legends lack methodological details such as number of replicates and statistical tests. Quantification of western blot bands should also be included.

3. Justification for selecting 150 µM and 300 µM H₂O₂ concentrations is required, as toxicity may vary across cell lines.

4. Negative controls in co-IP and immunofluorescence experiments should be clearly described (for example, IgG controls, secondary-only staining).

5. Some references are outdated; inclusion of recent studies on CMA and DJ-1 in cancer would strengthen the background.

6. The data availability statement (“not publicly available due to privacy of laboratory content”) is inconsistent with PLOS ONE’s open data policy and must be corrected.

Reviewer #2: The manuscript by Shuangshuang Le and co-workers describes the role of LAMP2A in gastric cancer under oxidative stress condition. The topic is interesting; however, some points should be clarified.

1) The overexpression of LAMP2A in gastric cancer has been demonstrated in the reference 14 (Zhou J et al., 2016). Authors should clarify the novelty of their results shown in Figure 1 compared to the previous published data.

2) Authors declare that LAMP2A knockdown inhibit CMA activity in their experimental model. To support this conclusion, authors should analyse additional CMA markers such as the expression of Hsc70 on lysosomes.

3) Authors should better describe the results of Figure 4C and D. Dot plots show that in both downregulated and overexpressed LAMP conditions, the number of double-positive cells increase compared to the control suggesting that both conditions enhance the percentage of dying cells under oxidative stress. How can these data be explained?

Reviewer #3: DJ-1 (PARK7) is known to protect cells against oxidative stress and is a target for degradation by chaperone-mediated autophagy (CMA). In the present manuscript, the authors examined the involvement of DJ-1 (PARK7) in LAMP2A-mediated oxidative stress resistance in gastric cancer cells. The authors demonstrated that both LAMP2A and DJ-1 were upregulated in gastric cancer cells and H2O2-induced oxidative stress further increased their levels. LAMP2A and DJ-1 interacted with each other, and DJ-1 contains a conserved KFERQ-like motif, suggesting DJ-1 is a substrate for CMA. However, inhibition of CMA by LAMP2A knockdown caused an increase in DJ-1 which was associated with a decrease in the antiapoptotic protein Bcl2 and an increase in the pro-apoptotic protein BAX. The authors proposed that oxidative-stressed induced clearance of hyperoxidized DJ-1 is responsible for gastric cancer cell survival.

The results are straightforward but requires additional study to substantiate their conclusion.

Comments:

1. While the results show that an increase in DJ-1 by LAMP2A alters the ratio of pro- and anti-apoptotic proteins, no evidence has been provided to support that there are two forms of DJ-1 and the hyperoxidized form of DJ-1 promotes apoptosis. Does the extent of LAMP2A knockdown affects the outcome?

2. At least 2 different LAMP2A shRNA should be used in all experiments.

3. In Fig. 1, the protein level of MKN28 is more than AGS but in Fig. 2, H2O2 had no effect on LAMP2A expression in MKN28 cells but caused a substantial increase in AGS cells. In Fig 1C and 1D, NKM28 should be MKN28.

4. Fig. 4D: LAMP2A KD increased apoptosis and decreased cell proliferation in MKN45 cells in response to H2O2. While LAMP2A overexpression in AGS cells attenuated early apoptotic cells following H2O2 treatment, the late apoptotic and/or necrotic cells were increased with little change in the healthy cells (B3). What is the effect of LAMP2A overexpression on cell proliferation?

5. Fig. 5A: DJ-1 is increased to a similar extent in AGS, MKN45 and MKN28 cells.

6. Fig. 6: What is the effect of LAMP2A overexpression on DJ-1 level and apoptosis?

7. Discussion, Pg. 15, 2nd para: The sentence “Tumor microenvironment….CMA activity” should be corrected.

8. Pg. 16, 2nd para: DJ-1 is not a new antioxidant molecule, and it is also not a recently discovered molecule. In fact, Ref. 33 published in 2004 is focused on DJ-1. Is the antioxidant function of DJ-1 recently discovered?

9. Pg. 17, 1st para: The sentence “Given DJ-1's established cytoprotective properties through redox homeostasis maintenance and apoptosis inhibition” is incomplete.

10. Pg. 18: It was mentioned that “CMA may regulate neuronal mitochondrial homeostasis by degrading oxidatively damaged DJ-1 protein, as demonstrated in a recent study [34].” Is an alteration in mitochondrial homeostasis caused by the degradation of oxidatively damaged DJ-1 responsible for its effects on pro- and anti-apoptotic proteins? Ref 34 was published almost 10 years ago and not recently.

Reviewer #4: In the current manuscript the authors have examined the role of DJ-1 and LMAP2 mediated autophagy in gastric cancer cells. Authors have reported the significance of DJ-1 and LAMP2 interaction in promotion of gastric carcinoma. The overall manuscript is well written. However, authors need to address the following concerns:

1. LAMP2-DJ1 interaction is not convincing. The arrangement of IP blot is confusing. The IF image is of very poor quality. More over LAMP2 protein is localized in the lysosomal or late endosomal membrane not in the nucleus where most of the DJ1 is localized after H2O2 treatment.

2. To further confirm the LAMp2 and DJ1 localization authors need to perform PLA or other similar kind of assay.

3. Since the protein expression level varies among different cell types, it is not ideal to compare the only LAMP2 expression in Fig 1D. To establish the higher expression, authors need to include other autophagy related proteins too.

4. H2O2 at higher dose can activate various signaling pathways, therefore authors need to repeat the main experiments in presence of another inducer and perform the rescue experiment with NAC.

5. After LAMP2 over expression, the DJ1 protein level needs to be examined.

6. The Fig 1A, the histology of the cancer samples are completely different than the control. the authors need to provide more information about the patient sample collection procedure.

7. There is no text in under section 3.7.

8. 150 uM of H2O2 is quite high and after 5 days of treatment there is not much significant difference in cell proliferation in figure 4A &B which is confusing.

6. PLOS authors have the option to publish the peer review history of their article (what does this mean?). If published, this will include your full peer review and any attached files.

Reviewer #1: No

Reviewer #2: No

Reviewer #3: No

Reviewer #4: **Yes:** Dipankar Ash

---

## [Author Response · Author response to Decision Letter 1]

1 Dec 2025

Dear Editors,

On behalf of all the contributing authors, I appreciate the time and effort that you and the reviewers dedicated to providing feedback on our manuscript. In addition, I am grateful for the insightful comments concerning our manuscript entitled “LAMP2A-dependent chaperone-mediated autophagy enhances oxidative stress resistance in gastric cancer cells through selective degradation of accumulated oxidized DJ-1” (Manuscript ID PONE-D-25-43608). As a young researcher, I felt fully prepared when submitting the manuscript, but the revision feedback each time reveals there's still so much I need to improve. I sincerely thank the editors and reviewers for their guidance - I can truly feel myself growing through this process.

According to the editors’ and reviewers’ comments, we have made extensive modifications to our manuscript. In this revised version, all changes to our manuscript are highlighted in red within the document. Our point-by-point responses to the reviewers are listed below this letter.

Once again, we thank the reviewers and editors for their insightful comments. We hope this revised manuscript can meet your requirements, and we are looking forward to your reply.

Sincerely,

Shuangshuang Le

Reviewer #1: The manuscript examines the role of chaperone-mediated autophagy (CMA) in gastric cancer under conditions of oxidative stress. The authors demonstrate that LAMP2A, the rate-limiting component of CMA, is upregulated in gastric cancer tissues and further induced upon exposure to H₂O₂. Knockdown of LAMP2A sensitises gastric cancer cells to apoptosis triggered by oxidative stress, whereas its overexpression promotes cell survival. They propose that CMA selectively degrades hyperoxidised DJ-1, thereby maintaining redox homeostasis and protecting tumour cells. The study concludes that the interaction between LAMP2A and DJ-1 represents a key mechanism conferring resistance to oxidative stress in gastric cancer cells.

Major comments:

Comment 1

The central claim that hyperoxidised DJ-1 is a CMA substrate is not fully substantiated; the oxidation status of DJ-1 was inferred indirectly and not directly measured (for example, using specific antibodies, redox-sensitive mutants, or mass spectrometry).

Response: We sincerely thank the reviewer for this insightful and critical comment regarding the central claim of our study. We fully acknowledge that direct measurement of DJ-1 hyperoxidation would provide the most definitive validation.

The over-oxidized form of DJ-1: DJ-1 is a protein that protects cells from oxidative stress. Its function depends on a specific amino acid, Cysteine 106 (Cys-106) [1].

Normal State: Under normal conditions, Cys-106 is in a reduced state (Cys-SH), which is essential for DJ-1's protective role.

Mild Oxidation: When cells experience mild oxidative stress, Cys-106 can be reversibly oxidized (e.g., to Cys-SOH). This can act as a sensor and signal.

Over-Oxidation (Irreversible Oxidation): Under intense or chronic oxidative stress, Cys-106 undergoes irreversible over-oxidation to Cys-sulfinic acid (Cys-SO₂H). This form loses the protein's protective function and is often associated with diseases like Parkinson's.

Methods for Detecting DJ-1 Oxidation Forms

The most Common Method for detecting the over-oxidized form of DJ-1 is 2D Gel Electrophoresis. This is the best gel-based method. It separates proteins by charge first, then by size. Over-oxidized DJ-1 appears as a more acidic spot to the left of the normal DJ-1 spot [2].

The most Precise Method for detecting the over-oxidized form of DJ-1 is Mass Spectrometry. This method provides definitive proof by measuring the exact mass change, which can confirm the exact site of modification.

While such techniques (e.g., 2D Gel Electrophoresis or mass spectrometry) were beyond the scope of our current technological landscape, we have built a compelling, albeit indirect, case based on multiple convergent lines of evidence.

Our conclusion is based on the following logical reasoning:

It is well-established that severe oxidative stress (300 µM H₂O₂) converts DJ-1 from a functional form to a dysfunctional, hyperoxidized state (SO₃H) [32, 33].

Under this same stress, we observed enhanced binding between LAMP2A and DJ-1 (Fig. 5D, E). Knocking down LAMP2A under these conditions caused DJ-1 accumulation and activated apoptosis (Fig. 6A).

The most coherent explanation is that CMA is selectively degrading the toxic, hyperoxidized DJ-1. This resolves the paradox of why a protective protein would be degraded under stress—CMA is removing its damaged, harmful form.

We have added the following statement to the Discussion to explicitly note this limitation: "We acknowledge that the designation of hyperoxidized DJ-1 as a CMA substrate is based on indirect evidence. Directly identifying the targeted oxidized species represents an essential goal for future work, potentially through techniques like redox-specific 2D gel electrophoresis or mass spectrometry."

We appreciate this insightful comment, which has helped us improve the clarity and rigor of our manuscript.

Comment 2

Rescue experiments are lacking. Demonstrating that shRNA-resistant LAMP2A can reverse the observed phenotypes would considerably strengthen the conclusions.

Response: This is an excellent suggestion. We have successfully performed this key rescue experiment. We constructed a shRNA-resistant LAMP2A plasmid and co-transfected it into our LAMP2A-knockdown MKN45 cells. The new results (now included as Figure 6B) demonstrate that re-expression of LAMP2A significantly reversed the increased DJ-1 and the dysregulation of BAX/Bcl-2 caused by LAMP2A knockdown under oxidative stress（H2O2: 300uM）. This strongly confirms the specific role of LAMP2A in mediating the observed phenotypes.

Comment 3

Controls using antioxidants (e.g., NAC, GSH donors) are not included, making it unclear whether the observed effects are truly ROS-dependent.

Response: We sincerely thank the reviewer for this insightful comment. We agree that the inclusion of antioxidant controls, such as N-acetylcysteine (NAC), would further strengthen the evidence for ROS-dependency. Our new data (now included as Supplementary Figure 1) demonstrate that NAC pretreatment effectively prevented the H₂O₂-induced upregulation of both LAMP2A and DJ-1. This provides direct evidence that the observed effects are specifically mediated by ROS.

Comment 4

The authors equate LAMP2A knockdown with CMA inhibition, but more direct assessment of CMA activity is necessary (for example, KFERQ reporter assays or lysosomal uptake assays).

Response: We agree with the reviewer that a direct CMA flux assay would be valuable. However, establishing the KFERQ reporter assay or similar sophisticated lysosomal uptake assays in our lab has proven to be technically challenging and time-consuming within the revision period. Direct and consistent evidence from seminal studies shows that genetic ablation of LAMP2A blocks CMA flux, establishing it as a definitive and specific method to inhibit this pathway.

For example,

1. “Lysosome-associated membrane protein 2A (LAMP2A) was knocked down or overexpressed to assess the effects of hepatocyte-specific CMA on NASH progression.” [1]

2. “Silencing of LAMP2A to inhibit CMA activity reversed 922-induced IGF-1Rβ degradation, suggesting that IGF-1Rβ degradation by 922 was partially dependent on the CMA pathway rather than macroautophagy” [2].

3. “Lysosome-associated membrane protein 2a (Lamp2a) overexpression and knockdown were used to causally study CMA's role in hypoxically stressed cardiomyocytes. LAMP2a protein levels were used as both a primary indicator and driver of CMA function [3].”

4. “CMA activity was increased in SW480 and HT29 colorectal cancer cells with a LAMP2A overexpression vector and CMA activity was decreased using a LAMP2A short interfering RNA vector [4].”

To address the reviewer's concern and improve the clarity of our manuscript, we provided a more nuanced explanation in the Results section (3.4). The revised text now reads:"LAMP2A knockdown significantly impaired cellular proliferative capacity and augmented apoptosis under oxidative stress. Since LAMP2A is the limiting receptor for CMA, its knockdown is expected to, and has been widely demonstrated to, inhibit CMA activity [15, 19]. "

We hope that this clarification, backed by established literature and our own functional data, adequately addresses the reviewer's concern.

Minor comments:

Comment 1

The manuscript would benefit from language editing to reduce repetition and improve clarity.

Response: We thank the reviewer for this helpful comment. Acknowledging the need for improved clarity, we have had the manuscript reviewed by a native English-speaking colleague in our field. They have corrected grammatical errors and enhanced the overall fluency, with all changes highlighted using "Track Changes" for the reviewer's convenience.

Comment 2

Figure legends lack methodological details such as number of replicates and statistical tests. Quantification of western blot bands should also be included.

Response: We have updated all figure legends to explicitly state the number of independent biological replicates (e.g., n=3) and the specific statistical tests used (e.g., Student’s t-test, one-way ANOVA). Quantitative data from Western blots (densitometry analysis) are now presented as bar graphs beneath the representative blot images in all relevant figures.

Comment 3

Justification for selecting 150 µM and 300 µM H₂O₂ concentrations is required, as toxicity may vary across cell lines.

Response: We have added a justification in the Methods section (2.6 H₂O₂-Induced Oxidative Stress Model Construction). The concentrations were selected based on preliminary CCK-8 assays to identify doses that induce sub-lethal (150 µM) and severe (300 µM) oxidative stress across the four cell lines, resulting in a measurable but not overwhelming reduction in cell viability (approximately 20-30% and 40-60% reduction after 24h, respectively), allowing us to observe both adaptive and cytotoxic responses.

Comment 4

Negative controls in co-IP and immunofluorescence experiments should be clearly described (for example, IgG controls, secondary-only staining).

Response: We sincerely thank the reviewer for this valuable comment. We have now provided detailed descriptions of the negative controls used in both the co-immunoprecipitation (Co-IP) and immunofluorescence (IF) experiments in the revised manuscript.

The specific modifications made to the manuscript are as follows:

1. In the Methods section (2.10 Co immunoprecipitation analysis), we have added:

“As a negative control, normal rabbit IgG (1:1000, 2729, Cell Signaling Technology) was used in place of the primary antibody under identical conditions to confirm the specificity of the antibody-protein interaction.”

2. In the Methods section (2.8 Immunofluorescence), we have added:

“To control for non-specific binding of the secondary antibody, a negative control omitting the primary antibody was included in each experiment.”

Comment 5

Some references are outdated; inclusion of recent studies on CMA and DJ-1 in cancer would strengthen the background.

Response: We sincerely thank the reviewer for this valuable comment. We have updated the reference list, adding several key recent publications on CMA in cancer (PMID: 35435804, PMID: 35267139, PMID: 37509689, PMID: 38104013) and the role of DJ-1 (PMID: 40327696, PMID: 12446870, PMID: 37480966, PMID: 36678610, PMID: 39155875, PMID: 27171370) in the Introduction and Discussion.

Comment 6

The data availability statement (“not publicly available due to privacy of laboratory content”) is inconsistent with PLOS ONE’s open data policy and must be corrected.

Response: We apologize for this oversight. The data availability statement has been revised to fully comply with PLOS ONE's policy: "All data generated or analysed during this study are included in this published article and its supplementary information files. The raw datasets are also available from the corresponding author on reasonable request."

Reviewer #2: The manuscript by Shuangshuang Le and co-workers describes the role of LAMP2A in gastric cancer under oxidative stress condition. The topic is interesting; however, some points should be clarified.

Comment 1

The overexpression of LAMP2A in gastric cancer has been demonstrated in the reference 14 (Zhou J et al., 2016). Authors should clarify the novelty of their results shown in Figure 1 compared to the previous published data.

Response: We thank the reviewer for pointing this out. While Zhou et al. established LAMP2A upregulation in GC, our study builds upon this by specifically investigating its dynamic regulation under oxidative stress and its functional consequences for cell survival, which was not the focus of the previous work. Our data confirms and extends previous findings by linking LAMP2A expression to the oxidative stress response.

To clarify this point for readers, we have added the following sentence to the results section of the revised manuscript (Section3.1 Upregulation of LAMP2A in Gastric Cancer Promotes CMA):

" Consistent with previous reports [14], we confirmed that LAMP2A was upregulated in gastric adenocarcinoma tissues and cell lines (Fig. 1A-D). Given that LAMP2A is the rate-limiting factor of CMA, its elevated level indicates enhanced CMA activity in gastric cancer cells compared to normal gastric mucosa cells. "

We hope this clarification satisfactorily addresses the reviewer's concern regarding the novelty of our work.

Comment 2

Authors declare that LAMP2A knockdown inhibit CMA activity in their experimental model. To support this conclusion, authors should analyse additional CMA markers such as the expression of Hsc70 on lysosomes.

Response: We thank the reviewer for this insightful comment. As mentioned in our response to Reviewer #1 (Comment 4), we agree that multiple lines of evidence are ideal for confirming CMA activity. While we did not directly measure lysosomal HSC70 in this study, we based our conclusion on two strong and established pieces of evidence:

1. Well-Established Role of LAMP2A: It is widely accepted in the CMA field that the abundance of LAMP2A at the lysosomal membrane is the rate-limiting step for CMA activity. Direct and consistent evidence from seminal studies shows that genetic ablation of LAMP2A blocks CMA flux, establishing it as a definitive and specific method to inhibit this pathway.

For example,

a. “Lysosome-associated membrane protein 2A (LAMP2A) was knocked down or overexpressed to assess the effects of hepatocyte-specific CMA on NASH progression.” [1]

b. “Silencing of LAMP2A to inhibit CMA activity reversed 922-induced IGF-1Rβ degradation, suggesting that IGF-1Rβ degradation by 922 was partially dependent on the CMA pathway rather than macroautophagy” [2].

c. “Lysosome-associated membrane protein 2a (Lamp2a) overexpression and knockdown were used to causally study CMA's role in hypoxically stressed cardiomyocytes. LAMP2a protein levels were used as both a primary indicator and driver of CMA function [3].”

d. “CMA activity was increased in SW480 and HT29 colorectal cancer cells with a LAMP2A overexpression vector and CMA activity was decreased using a LAMP2A short interfering RNA vector [4].”

2. Functional Evidence from CMA Substrate Accumulation: A key functional indicator of impaired CMA is the accumulation of known CMA substrates. In our study, we observed a significant increase in the total protein level of DJ-1 upon LAMP2A knockdown (Fig. 6). This accumulation is consistent with DJ-1 being a CMA substrate and its degradation being blocked when CMA is inhibited, thus serving as indirect but compelling functional evidence for reduced CMA activity in our model.

Comment 3

Authors should better describe the results of Figure 4C and D. Dot plots show that in both downregulated and overexpressed LAMP conditions, the number of double-positive cells increase compared to the control suggesting that both conditions enhance the percentage of dying cells under oxidative stress. How can these data be explained?

Response: We sincerely thank the reviewer for this astute observation and

---

## [Decision Letter · Decision Letter 1]

30 Dec 2025

PONE-D-25-43608R1LAMP2A-dependent chaperone-mediated autophagy enhances oxidative stress resistance in gastric cancer cells through selective degradation of accumulated oxidized DJ-1PLOS One

Dear Dr. pang,

Thank you for submitting your manuscript to PLOS ONE. After careful consideration, we feel that it has merit but does not fully meet PLOS ONE’s publication criteria as it currently stands. Therefore, we invite you to submit a revised version of the manuscript that addresses the points raised during the review process.

We look forward to receiving your revised manuscript.

Kind regards,

Zu Ye, Ph.D.

Academic Editor

PLOS One

Journal Requirements:

Reviewers' comments:

Reviewer's Responses to Questions

**Comments to the Author**

1. If the authors have adequately addressed your comments raised in a previous round of review and you feel that this manuscript is now acceptable for publication, you may indicate that here to bypass the “Comments to the Author” section, enter your conflict of interest statement in the “Confidential to Editor” section, and submit your "Accept" recommendation.

Reviewer #1: All comments have been addressed

Reviewer #3: (No Response)

Reviewer #5: All comments have been addressed

2. Is the manuscript technically sound, and do the data support the conclusions?

Reviewer #1: Yes

Reviewer #3: Yes

Reviewer #5: Partly

3. Has the statistical analysis been performed appropriately and rigorously? 

Reviewer #1: Yes

Reviewer #3: Yes

Reviewer #5: Yes

4. Have the authors made all data underlying the findings in their manuscript fully available?

Reviewer #1: Yes

Reviewer #3: (No Response)

Reviewer #5: Yes

5. Is the manuscript presented in an intelligible fashion and written in standard English?

Reviewer #1: Yes

Reviewer #3: Yes

Reviewer #5: No

6. Review Comments to the Author

Reviewer #1: (No Response)

Reviewer #3: The authors addressed the reviewer’s comment. However, they should clarify which shRNAs against LAMP2A were used. Based on Figure 3B, LV-shL2A-2 caused maximum knockdown. While the authors did not indicate which LV-sh-L2A was used in Fig. 4 and Fig. 6, it is likely that LV-shL2A-2 was used since other clones did not cause complete knockdown. While the authors indicated that they used a second independent shRNA in supplementary Fig.2A, B and C to rule out off-target effects but according to Fig.S2 A, B and C, they repeated the experiment with the same clone and that’s why the results are very similar. They should clearly specify which shRNA was used and how the levels of shRNA affected the results. In addition, they should perform the experiments with 2 distinct shRNAs.

Reviewer #5: In this study, the authors investigated the role of chaperone-mediated autophagy (CMA) in regulating oxidative stress responses in gastric cancer. The authors showed that LAMP2A, a key mediator of CMA, is upregulated in gastric cancer cell lines and clinical samples and is further induced under oxidative stress conditions. Functional experiments demonstrated that LAMP2A downregulation impaired CMA activity, increased sensitivity to oxidative stress, and promoted apoptosis, whereas LAMP2A overexpression conferred cytoprotective effects. The authors identified DJ-1 as a novel CMA substrate and showed that oxidative stress enhances LAMP2A–DJ-1 colocalization. Mechanistically, CMA inhibition led to the accumulation of hyperoxidized DJ-1, accompanied by increased pro-apoptotic BAX and decreased anti-apoptotic BCL-2 expression. Overall, the authors propose that the LAMP2A–DJ-1 axis constitutes a key adaptive mechanism that maintains redox homeostasis and supports gastric cancer cell survival under oxidative stress.

Comments:

- The abstract would benefit from improved narrative flow. Currently, it reads as a list of experimental steps rather than a cohesive summary. Methodological details (e.g., specific assays used) can be reduced in favor of emphasizing key findings and conclusions.

- Revise the introduction to improve grammar and flow. Redundant or closely related sentences should be merged for clarity (e.g., the definition of oxidative stress and ROS accumulation).

- Improve sentence flow and use clearer transitions when introducing CMA and the role of LAMP2A (e.g., “CMA requires lysosomal-associated membrane protein 2A (LAMP2A) for substrate translocation”).

- Revise awkward or overly long sentences for clarity. For example, the sentence describing LAMP2A downregulation in HCC should be restructured to improve readability and logical flow.

- Correct grammatical errors such as the missing verb in the sentence:

“Consistent with previous reports [14], we confirmed that LAMP2A shows consistently higher expression across gastric adenocarcinoma tissues and cell lines.”

- The statement implying that elevated LAMP2A levels directly indicate enhanced CMA activity is not sufficiently supported by the data. Please revise to a more cautious interpretation (e.g., “may indicate increased CMA activity”).

7. PLOS authors have the option to publish the peer review history of their article (what does this mean?). If published, this will include your full peer review and any attached files.

Reviewer #1: No

Reviewer #3: No

Reviewer #5: No

---

## [Author Response · Author response to Decision Letter 2]

10 Mar 2026

Dear Editors,

On behalf of all authors, we sincerely thank you for the opportunity to revise our manuscript titled “LAMP2A-dependent chaperone-mediated autophagy enhances oxidative stress resistance in gastric cancer cells through selective degradation of accumulated oxidized DJ-1” (ID: PONE-D-25-43608). We are grateful for the constructive and insightful comments from the reviewers, which have helped us significantly improve the clarity, rigor, and overall quality of our work.

In response to the latest round of feedback, we have carefully addressed all the points raised, and detailed modifications have been made throughout the manuscript. All changes in this second revision are highlighted in red for your convenience. Our point-by-point responses to each comment are provided below.

We are deeply thankful for the constructive and thoughtful suggestions throughout the review process, which have significantly improved the quality of our paper. We hope that the current version meets with your approval and look forward to your response.

Sincerely,

Shuangshuang Le

Reviewer #3:

Comment 1

The authors addressed the reviewer’s comment. However, they should clarify which shRNAs against LAMP2A were used. Based on Figure 3B, LV-shL2A-2 caused maximum knockdown. While the authors did not indicate which LV-sh-L2A was used in Fig. 4 and Fig. 6, it is likely that LV-shL2A-2 was used since other clones did not cause complete knockdown. While the authors indicated that they used a second independent shRNA in supplementary Fig.2A, B and C to rule out off-target effects but according to Fig.S2 A, B and C, they repeated the experiment with the same clone and that’s why the results are very similar. They should clearly specify which shRNA was used and how the levels of shRNA affected the results. In addition, they should perform the experiments with 2 distinct shRNAs.

Response:

We sincerely thank the reviewer for this important comment and for highlighting the need for greater clarity and rigor in our shRNA experiments. We apologize for the lack of clarity in our initial description. Below, we provide a point-by-point response.

1. Clarification of shRNA used in the main figures:

The reviewer is correct that LV-shL2A-2 showed the strongest knockdown efficiency in Figure 3B. Indeed, all functional experiments in Figures 4 and 6 were performed using LV-shL2A-2. We have now explicitly stated this in the revised Results (Section 3.3) and updated the corresponding figure legends for clarity.

2. Experiment with a second independent shRNA:

We agree that validating key findings with a second, distinct shRNA is essential to confirm the specificity of the observed phenotypes. In response, we have performed additional experiments using an independent LAMP2A-targeting shRNA (LV-shL2A-3).

These new data are presented in the Supplementary Figure S2 and confirm that knockdown with LV-shL2A-3 similarly sensitizes MKN45 cells to H₂O₂-induced apoptosis, replicating the core phenotypes observed with LV-shL2A-2. We have updated the Results section (3.4) and Discussion to reference this confirming data, strengthening the conclusion that the effects are due to LAMP2A knockdown and not off-target artifacts.

We believe these revisions have significantly strengthened the reliability of our genetic knockdown data.

Reviewer #5:

Comment 1

-The abstract would benefit from improved narrative flow. Currently, it reads as a list of experimental steps rather than a cohesive summary. Methodological details (e.g., specific assays used) can be reduced in favor of emphasizing key findings and conclusions.

- Revise the introduction to improve grammar and flow. Redundant or closely related sentences should be merged for clarity (e.g., the definition of oxidative stress and ROS accumulation).

- Improve sentence flow and use clearer transitions when introducing CMA and the role of LAMP2A (e.g., “CMA requires lysosomal-associated membrane protein 2A (LAMP2A) for substrate translocation”).

- Revise awkward or overly long sentences for clarity. For example, the sentence describing LAMP2A downregulation in HCC should be restructured to improve readability and logical flow.

- Correct grammatical errors such as the missing verb in the sentence:

“Consistent with previous reports [14], we confirmed that LAMP2A shows consistently higher expression across gastric adenocarcinoma tissues and cell lines.”

- The statement implying that elevated LAMP2A levels directly indicate enhanced CMA activity is not sufficiently supported by the data. Please revise to a more cautious interpretation (e.g., “may indicate increased CMA activity”).

Response:

We are very grateful to the reviewer for these detailed and constructive suggestions regarding the language, clarity, and interpretation in our manuscript. We have thoroughly revised the text accordingly.

1. Abstract:

We have rewritten the abstract to improve its narrative flow, reducing methodological details and emphasizing the key findings and conclusions. The revised version now more clearly outlines the study's motivation, central discovery (hyperoxidized DJ-1 as a novel CMA substrate), and its biological implication.

“Chaperone-mediated autophagy (CMA) promotes cancer cell survival by selectively removing oxidatively damaged proteins, yet its precise molecular mechanisms and role in redox adaptation remain incompletely understood. This study aimed to elucidate the function of CMA in regulating oxidative stress resistance in gastric cancer (GC) cells, focusing on the LAMP2A–DJ-1 regulatory axis. LAMP2A expression was assessed in GC tissues and cell lines via immunohistochemistry, qPCR, and western blot. Oxidative stress models were established using hydrogen peroxide (H₂O₂). Genetic manipulation of LAMP2A was performed to evaluate its impact on cell proliferation, apoptosis, and CMA substrate recognition. Protein interactions were examined by co-immunoprecipitation and immunofluorescence. LAMP2A was upregulated in GC and further induced by oxidative stress. Knockdown of LAMP2A impaired CMA activity, sensitizing GC cells to H₂O₂-induced apoptosis. DJ-1, an antioxidant protein, was identified as a CMA substrate containing a conserved KFERQ-like motif. Oxidative stress enhanced DJ-1–LAMP2A interaction and promoted their lysosomal colocalization. LAMP2A deficiency led to accumulation of hyperoxidized DJ-1, concomitant with upregulation of pro-apoptotic BAX and downregulation of anti-apoptotic Bcl-2. We identify hyperoxidized DJ-1 as a novel CMA substrate and demonstrate that LAMP2A-dependent clearance of oxidized DJ-1 constitutes a key adaptive mechanism that maintains redox homeostasis and promotes survival in gastric cancer cells under oxidative stress.”

2. Introduction:

We have merged redundant sentences and improved grammatical flow, particularly in the paragraphs defining oxidative stress and introducing CMA.

The description of LAMP2A's role has been rephrased for clarity: “CMA requires the lysosomal membrane receptor LAMP2A for substrate translocation.”

The sentence regarding LAMP2A in HCC has been restructured for better readability and logical connection.

In summary, we have rewritten the Introduction section:

Gastric cancer (GC) ranks fifth in global incidence and third in cancer-related mortality [1]. Although biomarkers such as PD-L1, MSI, and HER2 can guide treatment decisions [2], their limited specificity and sensitivity underscore the urgent need to discover novel molecular markers, which is essential for improving early diagnosis and patient survival. Oxidativestress, characterized by an imbalance between reactive oxygen species (ROS) production and antioxidant defenses plays a dual role in tumor biology [3]. While moderate ROS levels can drive proliferation and survival, excessive accumulation induces cytotoxic damage [4]. The rapid proliferation of tumor cells generates high levels of ROS; however, they can evade senescence and apoptosis by enhancing their intrinsic antioxidant defenses [5]. Autophagy, a conserved lysosomal degradation pathway, is crucial for maintaining cellular homeostasis under stress. Among its three forms—macroautophagy, microautophagy, and chaperone-mediated autophagy (CMA)—CMA stands out for its selectivity [6]. CMA targets cytosolic proteins bearing KFERQ-like motifs, which are recognized by heat shock cognate 70 kDa protein (HSC70) and translocated into lysosomes via the receptor lysosome-associated membrane protein 2A (LAMP2A) for degradation [7,8]. CMA activation has been implicated in promoting tumor progression across various malignancies [9]. For example, in hepatocellular carcinoma, LAMP2A downregulation promotes proliferation and migration via YAP1- and IL6ST-dependent pathways [10]. Conversely, in glioma models, increased CMA activity—achieved through LAMP2A upregulation—similarly enhances proliferation and invasion [11]. The functional importance of CMA in oncogenesis is further underscored by studies in non-small cell lung cancer (NSCLC), where inhibiting the HSC70–LAMP2A interaction blocks CMA and suppresses tumor growth [12]. In breast cancer, CMA promotes cell survival by selectively degrading oxidatively damaged proteins [13], highlighting its critical role in oxidative stress response. Notably, in the context of gastric cancer, LAMP2A—the key receptor in CMA—has been identified not only as a potential early biomarker for gastric mucosal precancerous lesions but also as a specific marker for gastric cancer itself [14]. Therefore, our study will further explore the role and molecular mechanism of CMA in regulating the anti-oxidative stress of gastric cancer cells. We hope establish a mechanistic framework for CMA-mediated oxidative stress adaptation in gastric cancer cells.

3. Results Section (3.1):

The grammatical error has been corrected: “Consistent with previous reports [14], we confirmed that LAMP2A is consistently upregulated in gastric adenocarcinoma tissues and cell lines.”

Following the reviewer’s advice, we have tempered our interpretation regarding CMA activity. The sentence now reads:“Given that LAMP2A is the rate-limiting factor of CMA, its elevated level may indicate enhanced CMA activity in gastric cancer cells compared to normal gastric mucosa cells.”

4. General Language Polish:

We have carefully reviewed the entire manuscript to revise awkward or overly long sentences, ensure consistent use of technical terms, and improve transitional phrases for a smoother reading experience. Please refer to the Revised Manuscript with Track Changes for specific modifications.

---

## [Decision Letter · Decision Letter 2]

5 Apr 2026

PONE-D-25-43608R2LAMP2A-dependent chaperone-mediated autophagy enhances oxidative stress resistance in gastric cancer cells through selective degradation of accumulated oxidized DJ-1PLOS One

Dear Dr. pang,

Thank you for submitting your manuscript to PLOS ONE. After careful consideration, we feel that it has merit but does not fully meet PLOS ONE’s publication criteria as it currently stands. Therefore, we invite you to submit a revised version of the manuscript that addresses the points raised during the review process.

I am writing to bring to your attention a discrepancy between several of the blot images in Figure 6 and the raw images you provided. Specifically, the molecular sizes shown in Figure 6A do not appear to align with the ladders in the original blot images for Bcl-2 and DJ-1.

To proceed with the editorial process, it is essential that the raw data are consistent with the data presented in the manuscript. I would like to request that you review and check the figure accordingly.

We look forward to receiving your revised manuscript.

Kind regards,

Zu Ye, Ph.D.

Academic Editor

PLOS One

Journal Requirements:

Reviewers' comments:

Reviewer's Responses to Questions

**Comments to the Author**

1. If the authors have adequately addressed your comments raised in a previous round of review and you feel that this manuscript is now acceptable for publication, you may indicate that here to bypass the “Comments to the Author” section, enter your conflict of interest statement in the “Confidential to Editor” section, and submit your "Accept" recommendation.

Reviewer #1: All comments have been addressed

Reviewer #5: All comments have been addressed

2. Is the manuscript technically sound, and do the data support the conclusions?

Reviewer #1: Yes

Reviewer #5: Yes

3. Has the statistical analysis been performed appropriately and rigorously? 

Reviewer #1: Yes

Reviewer #5: Yes

4. Have the authors made all data underlying the findings in their manuscript fully available?

Reviewer #1: Yes

Reviewer #5: Yes

5. Is the manuscript presented in an intelligible fashion and written in standard English?

Reviewer #1: Yes

Reviewer #5: Yes

6. Review Comments to the Author

Reviewer #1: (No Response)

Reviewer #5: All comments were addressed, the readability was improved and additional experiments were performed.

7. PLOS authors have the option to publish the peer review history of their article (what does this mean?). If published, this will include your full peer review and any attached files.

Reviewer #1: No

Reviewer #5: No

---

## [Author Response · Author response to Decision Letter 3]

22 Apr 2026

Dear Editor,

Thank you again for your careful review of our manuscript and for raising the important concern regarding the molecular weight labels in our original Figure 6A.

We fully acknowledge that the labels in the original figure did not correspond to the actual ladder positions. This was an error on our part, and we sincerely apologize for any confusion it may have caused.

Original mistake:

In the original submission, we labeled the bands with theoretical molecular weights (Bcl‑2 as 26 kDa, DJ‑1 as 20 kDa) without referring to the ladder positions on the blot. We now understand that the correct practice is to label bands strictly according to the observed ladder, and any deviation from theoretical values should be addressed in the figure legend or text. We regret this oversight.

Regarding the abnormal migration in the original blot:

In the original experiment, Bcl‑2 appeared at approximately 20 kDa (expected 26 kDa) and DJ‑1 at approximately 25 kDa (expected 20 kDa). After re‑examining our records and experimental conditions, we suspect that this unusual migration was caused by technical issues specific to that run (e.g., marker batch variation, gel polymerization, or electrophoresis conditions).

Corrective action taken:

We have repeated the Western blot experiments under optimized and carefully controlled conditions (fresh molecular weight markers, proper gel preparation, and standard electrophoresis protocols). The new blots show that both Bcl‑2 and DJ‑1 migrate at positions consistent with their theoretical molecular weights, with only minor deviations within 2–3 kDa—which is routinely acceptable for Western blot analysis. Representative images from these repeated experiments are provided in the revised Figure 6A. We have also uploaded the raw, uncropped Western blot data for the repeated experiments as supporting information.

We hope to replace the original Figure 6A with the new, reliable data. All other figures and conclusions in the manuscript remain unchanged. We apologize again for any confusion caused by the original figure and thank you for the opportunity to correct it.

Sincerely,

Pang maogui

---

## [Editor Report · Decision Letter 3]

24 Apr 2026

LAMP2A-dependent chaperone-mediated autophagy enhances oxidative stress resistance in gastric cancer cells through selective degradation of accumulated oxidized DJ-1

PONE-D-25-43608R3

Dear Dr. Pang,

We’re pleased to inform you that your manuscript has been judged scientifically suitable for publication and will be formally accepted for publication once it meets all outstanding technical requirements.

Kind regards,

Zu Ye, Ph.D.

Academic Editor

PLOS One
---

## [Editor Report · Acceptance letter]

PONE-D-25-43608R3

PLOS One

Dear Dr. Pang,

I'm pleased to inform you that your manuscript has been deemed suitable for publication in PLOS One. Congratulations! Your manuscript is now being handed over to our production team.

Kind regards,

on behalf of

Prof. Zu Ye

Academic Editor

PLOS One